# Associating Increased Chemical Exposure to Hurricane Harvey in a Longitudinal Panel Using Silicone Wristbands

**DOI:** 10.3390/ijerph19116670

**Published:** 2022-05-30

**Authors:** Samantha M. Samon, Diana Rohlman, Lane G. Tidwell, Peter D. Hoffman, Abiodun O. Oluyomi, Kim A. Anderson

**Affiliations:** 1Department of Environmental & Molecular Toxicology, Oregon State University, Corvallis, OR 97331, USA; samonc@oregonstate.edu (S.M.S.); lane.tidwell@oregonstate.edu (L.G.T.); peter.hoffman@oregonstate.edu (P.D.H.); 2College of Public Health and Human Sciences, Oregon State University, Corvallis, OR 97331, USA; diana.rohlman@oregonstate.edu; 3Section of Epidemiology and Population Sciences, Department of Medicine, Baylor College of Medicine, Houston, TX 77030, USA; abiodun.oluyomi@bcm.edu; 4Gulf Coast Center for Precision Environmental Health, Baylor College of Medicine, Houston, TX 77030, USA

**Keywords:** Hurricane Harvey, passive sampling, disaster research, silicone wristbands

## Abstract

Hurricane Harvey was associated with flood-related damage to chemical plants and oil refineries, and the flooding of hazardous waste sites, including 13 Superfund sites. As clean-up efforts began, concerns were raised regarding the human health impact of possible increased chemical exposure resulting from the hurricane and subsequent flooding. Personal sampling devices in the form of silicone wristbands were deployed to a longitudinal panel of individuals *(n* = 99) within 45 days of the hurricane and again one year later in the Houston metropolitan area. Using gas chromatography–mass spectroscopy, each wristband was screened for 1500 chemicals and analyzed for 63 polycyclic aromatic hydrocarbons (PAHs). Chemical exposure levels found on the wristbands were generally higher post-Hurricane Harvey. In the 1500 screen, 188 chemicals were detected, 29 were detected in at least 30% of the study population, and of those, 79% (*n* = 23) were found in significantly higher concentrations (*p* < 0.05) post-Hurricane Harvey. Similarly, in PAH analysis, 51 chemicals were detected, 31 were detected in at least 30% of the study population, and 39% (*n* = 12) were found at statistically higher concentrations (*p* < 0.05) post-Hurricane Harvey. This study indicates that there were increased levels of chemical exposure after Hurricane Harvey in the Houston metropolitan area.

## 1. Introduction

Disasters are increasing in frequency throughout the globe and are becoming costlier and more unpredictable and catastrophic [1,2]. There is a rising number of incidents where disasters rooted in meteorological, geological, and hydrological mechanisms have inadvertently caused the release of hazardous materials associated with the industrial production of chemicals, and hence caused a secondary disaster to occur [3,4,5,6]. These compounded events, sometimes referred to as natural–technological disasters (NATECH events), could significantly impact the environment and the surrounding population [5]. During these scenarios, it is challenging yet necessary to evaluate population-level chemical exposure to inform policy decisions regarding the prevention of hazardous material release in future disasters, minimize potential health risks associated with the disaster scenario, and understand the ecological and human health repercussions if disasters where chemical exposure is a concern continue to occur.

Environmental sampling following a disaster typically occurs in close proximity to the release site and may not reflect human health exposure [7]. One alternative is personalized exposure assessment, which also allows for long-term health studies without relying on location and chemical-dispersion data. However, care should be taken in personal exposure studies following disasters, as intrusive sampling, such as the collection of multiple biological samples, may cause an undue burden to participants that are recovering from a disaster scenario [8]. Participant recruitment may suffer as a result [7,9]. Passive sampling techniques provide an alternative that can be used for post-disaster chemical exposure assessment at both release sites and as personalized samplers [10,11].

Passive sampling devices can be effectively used in disaster research, including personal passive sampling devices in the form of silicone wristbands. Silicone wristbands function by diffusing the vapor phase of volatile or semi-volatile organic compounds into a hydrophobic polymer [12,13]. While silicone wristbands do not represent the effective dose of or total exposure to a compound, the fraction of environmental chemicals that the wristband can sample forms a significant portion of human exposure via a combination of inhalation and dermal exposure [14,15,16]. Silicone wristbands were compared to biological samples, and numerous associations between the two were found, providing evidence that they are biomimetic and capture the fraction of available chemicals for human uptake [14,15,17,18,19]. A review of 39 studies using the silicone wristband found that they were successfully used to evaluate exposure to multiple chemical groups to include polycyclic aromatic hydrocarbons, flame retardants, pesticides in multiple countries and across multiple occupations [20]. Several studies reported associations between wristband concentrations and environmental exposures; for example, Paulik et al. found that individuals living close to unconventional natural-gas drilling had higher concentrations than those individuals living farther away [21]. In addition to capturing short- and long-term exposures [20], silicone wristbands are minimally invasive to participants, easy to use, and stable for transport in hot and cold conditions [17,20,22,23]. Additionally, they can be partnered with demographic and location data to determine associations between potential chemical exposure and health outcomes [24]. These properties render them ideal for assessing potential personal chemical exposure in a disaster scenario [11].

A recent example of a disaster where chemical exposure assessment was needed is Hurricane Harvey [3,25]. Hurricane Harvey made landfall on the Texas coast on 25 August, 2017 as a Category 4 tropical cyclone, and brought more than 50 inches of rainfall to some parts of the Houston metropolitan statistical area (MSA) [26]. In addition to the physical damage to infrastructure and buildings, Harvey was associated with flood-related damage to chemical plants and oil refineries, and the flooding of hazardous waste sites, including 13 Superfund sites. Houston MSA, which is the fifth most populous MSA in the U.S., is a major hub for the nation’s petrochemical [27,28] and plastics [26,29] industries. The lack of citywide zoning regulation and related urbanization exacerbated both flood risk [30,31,32,33] and potential exposure to environmental contaminants [28,34,35,36].

At the time when the hurricane took place, little was known about to which chemicals Houston residents were potentially exposed, although there were concerns about excess emissions from industrial centers and flooded superfund sites [37]. To capture a broad range of organic pollutants, silicone wristbands were deployed to individuals living within Houston MSA, and screened for 1500 volatile and semi-volatile organic pollutants (VOCs and SVOCs). There was additional concern about the impact of petrochemical facilities and polycyclic aromatic hydrocarbon (PAH) exposure. PAHs are ubiquitous environmental contaminants that stem from pyrogenic, petrogenic, or biogenic sources. The major source of PAHs is pyrogenic and results from the incomplete combustion of organic substances such as coal, oil, and wood, while petrogenic PAHs stem from petroleum products and are formed during crude oil maturation [38,39]. PAHs from both pyrogenic and petrogenic sources were expected to be present following Hurricane Harvey.

This study was designed to better understand individual exposures to organic pollutants following Hurricane Harvey, and to provide insight on personal chemical exposure assessment during a disaster. Specifically, this study aimed to utilize a longitudinal panel of participants with time points during Hurricane Harvey flooding and in a nondisaster scenario to determine if potential chemical exposure to a variety of SVOCs and PAHs was higher during the aftermath of Hurricane Harvey.

## 2. Materials and Methods

### 2.1. Study Design and Population

This analysis includes chemical-exposure data from the Houston Hurricane Harvey Health (Houston-3H) study originally discussed in Oluyomi et al. (2021) [40]. Briefly, the 3H project aims to better understand the link among Hurricane Harvey flooding, flood-related exposures, and health outcomes in Houston residents. The Houston-3H study was approved by the institutional review boards at Oregon State University, Baylor College of Medicine (BCM), and the University of Texas Health Science Center, and all participants provided informed consent.

The Houston-3H project consisted of multiple timepoints. The first sampling period hereafter referred to as “post-hurricane” began within one month of Hurricane Harvey, while parts of the Houston MSA were still experiencing flooding, and active flood clean-up was still occurring [40]. The second time point was taken approximately one year after the hurricane in non-disaster conditions and is hereafter referred to as the “estimated baseline”. At both the post-hurricane and estimated-baseline timepoints, participants wore silicone wristbands for seven days, and completed questionnaires related to their demographics and flood-related exposure. Participants were originally recruited from targeted flood-impacted neighborhoods (e.g., Addicks, Baytown, East Houston, and Bellaire-Meyerland) within the Houston metropolitan area in Harris County, TX. Recruitment for the estimated-baseline timepoint consisted of contacting participants from the original time point that had indicated interest in continuing on in the study, and new participants living within the original study neighborhoods were recruited. Inclusion criteria for the study were being at least five years of age and fluent in either English or Spanish, and having given informed consent. Multiple individuals from the same household were allowed to participate in the study. Wristbands were returned between 21 September and 20 October 2017 for the post-hurricane timepoint (*n* = 172), and between 13 September and 29 October 2018 for the estimated-baseline timepoint (*n* = 239) (Figure 1). There was a longitudinal panel of 99 participants who had returned wristbands at both time points.

### 2.2. Chemical Exposure Assessment

#### 2.2.1. Silicone-Wristband Preparation and Extraction

Conditioning, post-deployment cleaning, and wristband extraction were performed as previously described [22] with limited modifications. Silicone wristbands (size large—width: 1.3 cm; inner diameter: 6.0 cm), were purchased from 24hourwristbands.com (Houston, TX, USA). Upon receipt from the manufacturer, wristbands were rinsed in deionized water to remove particulate matter before being conditioned at 270–300 °C for 180 min under vacuum at 0.1 Torr (Vacuum Oven, Blue-M, model no. POM18VC-2, with Welch Duo-Seal pump, model no. 1405, Mt. Prospect, IL, USA). Wristbands were then stored in sealed metal containers at 4 °C. Prior to deployment, the wristbands were transferred to air-tight polytetrafluoroethylene (PTFE) bags that were used for storage before and after deployment.

Before the returned wristbands were extracted, they first underwent post-deployment cleaning: they were rinsed twice with 18 MΩ-cm water and once with isopropanol to remove particulate matter. Following post-deployment cleaning, extraction methods varied depending on the time point. All wristband samples collected from the post-hurricane timepoint were extracted as indicated in Dixon et al., 2019 [41]. Wristband samples collected during the estimated-baseline timepoint were cut in half using solvent-rinsed scissors, and each half was stored in amber glass jars at −20 °C: one half was archived, and the other extracted. Prior to extraction, samples were spiked with recovery surrogates. Wristbands were extracted twice with 50 mL of ethyl acetate at ambient temperature. Sample extracts were combined and reduced to 1 mL under nitrogen (TurboVap LV, Biotage, Charlotte, NC, USA; RapidVap, LabConco, Kansas City, MO, USA; N-EVAP 111, Organomation Associates, Berlin, MA, USA). Aliquots of 100 μL underwent solid-phase extraction (SPE) using 3.5 mL acetonitrile loaded onto C18 SPE cartridges (Cleanert S C18, Agela Technologies, Torrance, CA, USA). Aliquots were solvent-exchanged to iso-octane (OA-SYS N-EVAP 111, Organomation Associates, Berlin, MA, USA) and stored at 4 °C before instrument analysis. Laboratory blanks were utilized at every step of the process to ensure that the study met data quality objectives. These included a total laboratory processing blank, post-deployment cleaning blank, extraction blank, and SPE blanks.

#### 2.2.2. 1500 Screening Method

Extracts from silicone wristbands were screened for 1500 target analytes using an Agilent 6890 N gas chromatograph (GC) with a 5975B Mass Selective Detector in full scan mode in conjunction with a predictive model automated mass spectral deconvolution and identification system. This method quantifies target analytes within a factor of 2.5 of the actual value. Further details regarding the analytical method, including limits of quantification (LOQs), were reported (Bergmann et al., 2018). All calibration verifications met the data-quality objectives of within 2.5 times the true value for 60% of the target analytes. Background subtraction for the post-hurricane timepoint was performed on the basis of a laboratory processing blank, and the estimated-baseline timepoint did not undergo background subtraction, as no chemicals were detected in the laboratory processing blank (Appendix A).

Detected analytes from the 1500 screen were subdivided on the basis of previously published chemical use and type categorization [41,42,43]. Chemicals were first categorized by chemical class. Specifically, polychlorinated biphenyls (PCBs), polycyclic aromatic hydrocarbons (PAHs), phthalates, and dioxins/furans were acknowledged due to prevalence and/or toxicological concern. The remaining chemicals were sorted by primary use: these categories included personal care products (PCPs), pesticides, flame retardants, pharmacological compounds, and industrial compounds (i.e., chemicals that have many uses and did not fit in any other category). *Many chemicals could be organized into multiple categories, but for the purposes of the study, they were sorted into one primary use or type*. A full list of detected analytes and their chemical categorization can be found in Appendix A. The full target list includes 124 flame retardants, 185 industrial compounds, 98 PAHs, 773 pesticides, 76 personal care products, 14 phthalates, and 260 polychlorinated biphenyls (PCBs) including dioxins and furans. A complete list of target analytes can be found at http://fses.oregonstate.edu/1530 (accessed on 26 May 2022).

#### 2.2.3. PAH Analysis Method

In total, 63 parent and alkylated PAHs were quantified using an Agilent (Santa Clara, CA, USA) 7890 gas chromatograph (GC) with a 7000 triple–quadruple mass spectrometer (MS/MS) (Appendix A). For this method, we used an Agilent Select PAH column, and each PAH in the method was calibrated with a curve of at least five points (correlations ≥ 0.99). All calibration verifications met data quality objectives +/− 30% of the true value for 70% of the target analytes [12,14,21,22]. Background subtraction was performed using appropriate blanks showing the highest analyte concentrations (Appendix A). PAHs were categorized on the basis of ring size and parent versus alkylated status.

### 2.3. Statistical Analysis

Statistical analyses were performed using SAS software version 9.4 (SAS Institute Inc., Cary, NC, USA) and JMP Pro version 15.2.1 (SAS Institute Inc.). The concentrations of all chemicals were converted into nanomoles per wristband (nmol/WB). Concentrations below the instrument’s limits of detection (LOD) were substituted with a value equal to LOD/√2. Concentrations below the limit of quantification (LOQ) were unchanged. To limit analysis to common chemical exposures, only compounds detected in at least 30% of the population in at least one time point were analyzed for both the 1500 screen and the PAH method. Additionally, to limit redundancy in analysis, PAHs in both the 1500 screen and 63 PAH method were excluded from statistical analysis relating to the 1500 screening method.

To leverage the multi-timepoint study design, a Wilcoxon matched-pair signed-rank test was utilized for individual analytes and various chemical groupings (e.g., all detected chemicals, chemical use and type categories, PAH ring size, and PAH substitution). Chemical groupings represent summed concentration (Σ[group]) for all analytes detected within that designation regardless of detection frequency. All chemical groups from both the 1500 screening method and 63 PAH were investigated for associations with questionnaire data, including the area deprivation index (ADI) [40], study neighborhood, participant race or ethnicity, age, gender, and home flooding status. Quartiles utilized to segregate ADI data were derived by using the entire post-hurricane study population (Samon et al., 2022). Univariate analysis consisting of a Kruskal–Wallis or Mann–Whitney U was conducted using the differences in chemical concentration between timepoints as the dependent variable. Differences were calculated by subtracting individual estimated-baseline concentrations from the paired post-hurricane concentration. A Bonferroni correction was conducted for covariate analysis. The new *p*-value was the alpha value (α_original_ = 0.05) divided by the number of comparisons (6): (α_altered_ = 0.05/6) = 0.008. To determine if any of the six correlations were statistically significant, the *p*-value had to be *p* < 0.008.

## 3. Results

### 3.1. Participant Characteristics

Participant characteristics are described in Appendix A. All reported characteristics are from questionnaires completed during the post-hurricane timepoint. Questionnaire data were generally not associated with any chemical grouping with one exception following the Bonferroni correction (Appendix A). Participant age was associated with Σ[flame retardants] (*p* = 0.006) (Appendix A).

### 3.2. 1500 Screen Results

Within the longitudinal panel of the participants, 162 different chemicals were detected post-hurricane, 137 were detected at the estimated baseline, and 101 chemicals were common across both time points. Overall, there were 188 unique chemicals detected across both time points. Post-hurricane, wristbands had a mean of 26 chemical detections, ranging from a minimum of 6 to a maximum of 43. Conversely, wristbands collected at the estimated baseline had a mean of 25 chemical detections, ranging from a minimum of 14 to a maximum of 40. The sum concentration of all detected analytes was statistically higher at the post-hurricane timepoint (*p* = 0.0034) (Appendix A). There were 26 compounds detected in at least 30% of the study population post-hurricane, 25 at the estimated baseline, and 22 chemicals had a greater than 30% detection frequency for both timepoints (Table 1).

#### 3.2.1. 1500 Screen Chemical Categories

Out of the 162 unique chemicals detected post-hurricane, there were 36 PCPs, 41 pesticides, 14 flame retardants, two pharmacological compounds, four PCBs, 25 PAHs, 10 phthalates, and 30 industrial compounds. Alternatively, the 137 individual chemicals detected at the estimated baseline contained 31 PCPs, 32 pesticides, 11 flame retardants, two pharmacological compounds, one PCB, 14 PAHs, 11 phthalates, one dioxin/furan, and 34 industrial compounds. At both time points, PCPs and phthalates formed the majority of the average total detected chemical exposure. PCPs were the predominant chemical category present post-hurricane, and phthalates were the predominant chemical category at the estimated baseline (Appendix A).

Σ[PCPs], Σ[pesticides], Σ[industrial], Σ[flame retardants], and Σ[pharmacological] were all found in statistically higher concentrations post-hurricane (*p* < 0.05) (Figure 2, Appendix A). Σ[phthalates] were found in higher concentrations at the estimated baseline, but this finding was not statically significant. Analysis was not conducted on Σ[PAHs] detected in the screening method, Σ[Dioxins/furans], or Σ[PCBs]. To limit redundancy with the 63 PAH method, PAHs were excluded from statistical analysis relating to the 1500 screening method, and both dioxins/furans and PCBs had very low levels of detections. A dioxin/furan was only detected once across both time points, and PCBs were only detected six times across both timepoints in five discrete wristbands.

#### 3.2.2. Individual Chemical Analysis

On average, 75% of the chemicals detected across both timepoints had higher concentrations post-hurricane. Of the commonly detected chemicals (found in >30% of samples), 79% had statistically higher concentrations post-hurricane (Table 1). This included the 10 most frequently detected compounds across both time points: di-n-butyl phthalate (*p* < 0.0001), galaxolide (*p* = 0.016), butyl benzyl phthalate (*p* < 0.0001), diisobutyl phthalate (*p* < 0.0001), N,N-diethyl-m-toluamide (*p* = 0.0013), butylated hydroxytoluene (*p* < 0.0001), tonalide (*p* < 0.0001), lilial (*p* < 0.0001),benzophenone (*p* < 0.0001), and diethyl phthalate (*p* < 0.0001). Of the commonly detected chemicals analyzed in the 1500 screen, 10% had statistically higher concentrations at the estimated baseline. This included benzyl salicylate (*p =* 0.00078), bis(2-ethylhexyl) phthalate (*p* < 0.0001), and coumarin (*p* = 0.038) (Table 1). Lastly, the composition of individual chemicals differed across timepoints (Figure 3).

### 3.3. 63 PAH Results

Within the longitudinal cohort of the participants, 49 PAHs were detected post-hurricane, 47 PAHs were detected at the estimated baseline, and 45 PAHs were common across both time points. There were 51 unique PAHs detected across both timepoints. Of these, 31 were commonly detected (found in >30% of samples) post-hurricane, 21 were commonly detected at the estimated baseline, and 21 chemicals were commonly detected at both timepoints. The number of PAHs found per wristband post-hurricane ranged from a minimum of 9 to a maximum of 37, with a mean of 24. At the estimated baseline the number of PAHs detected per wristband ranged from 12 to 39 with a mean of 19. The sum concentration of all PAHs detected was statistically higher at the post-hurricane timepoint (*p* = 0.0393) compared to the estimated baseline.

#### 3.3.1. PAH Categories

PAHs were categorized on the basis of ring size and PAH substitution (i.e., whether a PAH was a parent compound or alkylated). Post-hurricane, 61% of the PAHs detected were parent PAHs, and 39% were alkylated. At the estimated baseline, 66% of the PAHs detected were parent PAHs, and 34% were alkylated. There were ten 2-ring PAHs, 12 3-ring PAHs, 12 4-rings PAHs, seven 5-ring PAHs, and eight 6&7-ring PAHs post-hurricane. At the estimated baseline, there were nine 2-ring PAHs, 11 3-ring PAHs, 11 4-ring PAHs, eight 5-ring PAHs, and eight 6&7-ring PAHs.

Σ[parent PAHs] was in statistically higher concentration at the post-hurricane timepoint (*p* = 0.000328), while there were no differences across timepoints for alkylated PAHs. When evaluated by ring size, Σ[3-ring PAHs], Σ[4-ring PAHs], Σ[5-ring PAHs], and Σ[6&7-ring PAHs] were all in statistically higher concentrations at the post-hurricane timepoint (*p* < 0.05) (Figure 4, Appendix A). The sum concentration of 2-ring PAHs was on average higher at the estimated baseline, but this finding was not statistically significant (Appendix A). The total average detected PAH exposure was primarily composed of 2-ring and 3-ring PAHs. The composition of detected PAHs did not change dramatically across timepoints, with limited exceptions. Naphthalene formed a greater proportion of the 2-ring PAHs at the estimated baseline, and benzo[ghi]perylene formed a greater proportion of the 6& 7-ring PAHs post-hurricane (Appendix A).

#### 3.3.2. Individual PAH Analysis

On average, 72% of the detected PAHs had higher concentrations post-hurricane. Of the commonly detected chemicals, 39% had statistically higher concentrations post-hurricane. This included three 3-ring PAHs (phenanthrene (*p* = 0.00142), acenaphthene (*p* < 0.0001), and 2,3-dimethyanthracene (*p* < 0.0001)) and two 4-ring PAHs (pyrene (*p* < 0.0001) and fluoranthene (*p* < 0.0001)). All commonly detected 5-, 6-, and 7-ring PAHs also had statistically higher concentrations at the post-hurricane timepoint: benzo[b]fluoranthene (*p* < 0.0001), benzo[e]pyrene (*p* < 0.0001), benzo[k]fluoranthene (*p* = 00.0003), benzo[j]fluoranthene (*p* = 0.0007), benzo[ghi]perylene (*p* < 0.0001), and coronene (*p* < 0.0001) (Table 2).

Two 2-ring PAHs and one 3-ring PAH had higher concentrations at the estimated baseline. These included naphthalene (*p* = 0.0021), 2-methlnaphthalene (*p* = 0.0132, and 3,6-dimethylphenanthrene (*p* < 0.0001) (Table 2).

## 4. Discussion

The results from this analysis indicate that Houston MSA residents were exposed to higher levels of VOCs and PAHs during the post-hurricane sampling period vs. the estimated baseline. This was true for the sum of all measured chemicals using the 1500 screen and 63 PAHs method, the sum of multiple chemical categories (i.e., pesticides, flame retardants, and pharmacological compounds), and most PAH ring sizes, in addition to individual chemicals. When sampling occurred post-hurricane, all major highways were open in the Houston MSA, and industrial production for plastic resin and organic chemicals, and oil refineries were at ≥90% normal production capacity [44]. Therefore, this study expected to capture chemical exposure associated with daily life in the Houston MSA in addition to chemical exposures associated with the hurricane at the post-hurricane timepoint. The estimated-baseline data showed endemic levels of chemical-exposure generally present in the Houston MSA. Additionally, covariates often associated with chemical exposure (i.e., age, gender, race/ethnicity, and ADI) were not significantly associated with the paired differences in chemical-exposure measures in the wristbands across timepoints. Rather, the predominant factor attributed to changes in chemical exposure appeared to be due to the hurricane, which caused significant flooding and industrial pollution [3,25,26,44].

The majority of detected chemicals (75%) in the 1500 screening method across both timepoints had on average higher concentrations post-hurricane. Of the commonly detected chemicals (found in >30% of samples), 79% had statistically higher concentrations post-hurricane. Most chemical categories were also found in higher concentrations post-hurricane. This included personal care products, pesticides, flame retardants, and pharmacological compounds. Only phthalates had on average higher levels at the estimated baseline, but this finding was not statistically significant, and the large proportion of bis(2-ethylhexyl) phthalate at the estimated baseline drove that finding. The chemical makeup of all of categorizations associated with the 1500 screening method changed to an extent between timepoints, which is indicative of different chemical sources at each time point.

Results showed that 3-ring PAHs, 4-ring PAHs, 5-ring PAHs, and 6&7-ring PAHs all had higher post-hurricane levels. When the most frequently detected PAHs were individually compared across timepoints, heavier-molecular-weight PAHs were identified as having higher concentrations post-hurricane. This is toxicologically relevant, as PAH toxicity generally increases with molecular weight [39,45]. Some PAHs are well-known as carcinogens, mutagens, and teratogens, and thereby pose a serious threat to the health and wellbeing of humans [39,46,47]. The most significant expected health effect from inhalation exposure to PAHs is an excess risk of cancer [39]. Among PAHs that had statistically higher concentrations post-hurricane, benzo[b]fluoranthene, benzo[j]fluoranthene, and benzo[k]fluoranthene were included in the National Toxicology Programs Report on Carcinogens, Fifteenth Edition [48]. The results reported here indicate that there was increased presence of carcinogenic PAHs following Hurricane Harvey.

### 4.1. Chemical Exposure Assessment during a Disaster

Exposure assessment following disasters is essential to evaluate possible health effects. Since the frequency of disasters is likely to increase [2], it is important to evaluate the success and failure of current disaster research approaches in exposure assessment for future implementation. The Houston-3H project collectively stands out in many ways. First, the Houston-3H project utilized community input to establish study goals in which the community was interested [11,40]. Engaging the community led to increased community interest, evidenced by increased participant enrollment in the estimated-baseline timepoint and admirable participant compliance in the range of 79–90% for all collected timepoints. In disaster studies, there are often no baseline data to which disaster-related exposure levels can be compared [7,8,49]. Given the lack of prior studies using similar technology in Houston, no baseline data were available. However, this study successfully demonstrated that representative baseline data could be taken when the disaster is mitigated, one year later during the same season. Lastly, the choice of sampling tool, namely, silicone wristbands, was well-suited to the study design. Silicone wristbands are minimally invasive and easy to use in disaster scenarios [11], provide individual potential exposure assessments, and are capable of rapidly collecting data. The ease of use also likely contributed to a high compliance rate among participants.

### 4.2. Limitations

While the study aimed to evaluate chemical exposure immediately after the hurricane, due to logistics, the samples were collected over a month after the Hurricane Harvey landfall. At that point in time, factors other than the hurricane could have contributed to chemical exposure, but distinct differences were still seen between timepoints. As is typical in multi-timepoint studies, some participant loss occurred, but this did not interfere with the power of analysis. This study utilized a 1500 screening method as a semiquantitative screen. While the screening method is capable of quantifying a large number of chemicals, it sacrifices sensitivity and accuracy [42]. Therefore, for further resolution, an additional PAH-specific analytical method was utilized. Lastly, silicone wristbands, the sampling technology utilized in this study, capture a combination of inhalation and dermal exposure. At this time, chemical exposures associated with a specific exposure pathway cannot be isolated. Values obtained with the silicone wristband cannot be compared to reference values or health guidelines [12,13,17]; additionally, while some such values exist, most chemicals assessed in this study lack established reference values or health guideline values.

## 5. Conclusions

In the last few years, multiple studies reported increased chemical exposure following Hurricane Harvey [34,35,36,50,51,52]. This is the first study to identify higher levels of chemical exposure using personal samples after Hurricane Harvey, and to utilize a longitudinal panel of participants to evaluate chemical exposure across timepoints in a disaster scenario. A major limitation of post-disaster chemical exposure assessment is often the lack of baseline data. In both epidemiological and risk assessment studies, baseline data are necessary to determine whether and how the disaster impacted exposure [9]. While it was not possible to obtain personal-exposure data from before the hurricane, this study successfully demonstrated the use of a post-disaster timepoint to act as an estimated baseline. This study design was further strengthened through the use of paired samples, which showed generally higher levels of chemicals in wristbands worn immediately following Hurricane Harvey.

## Figures and Tables

**Figure 1 ijerph-19-06670-f001:**
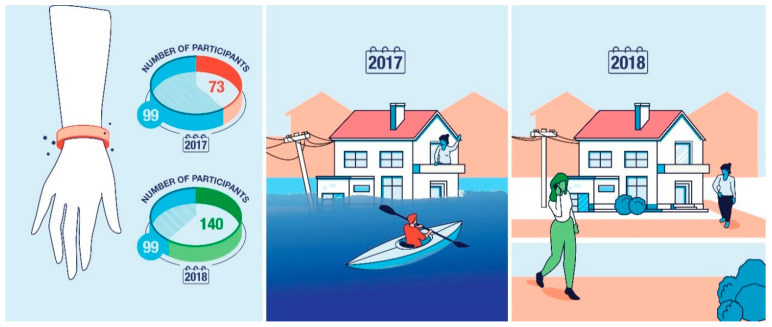
Participants wore and returned a silicone wristband during a post-hurricane timepoint in 2017 (n = 172, 83% compliance), and again one year later, in 2018 (n = 239, 90% compliance). The 2018 timepoint was utilized as an estimated baseline of chemical exposure in non-disaster conditions in Houston, TX. A longitudinal panel (n = 99, 79% compliance) participated in both the post-hurricane and estimated-baseline timepoints.

**Figure 2 ijerph-19-06670-f002:**
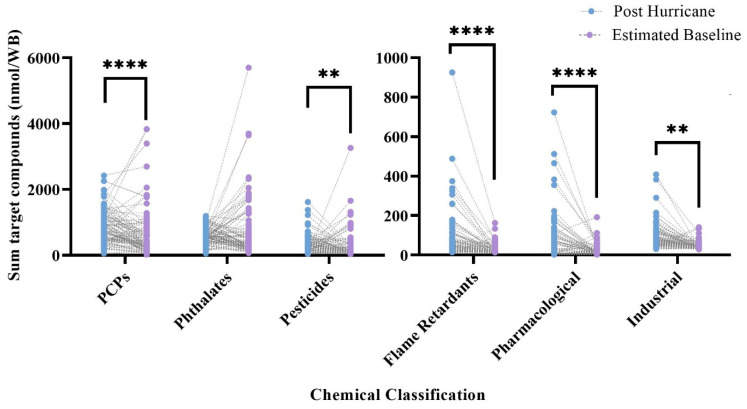
Sum concentration of chemical classifications (i.e., personal-care products (PCPs), phthalates, pesticides, flame retardants, and pharmacological and industrial products) for matched pairs across timepoints. Comparisons represent results from Wilcoxon matched-pair signed-rank tests ** (*p* < 0.01), **** (*p* < 0.0001).

**Figure 3 ijerph-19-06670-f003:**
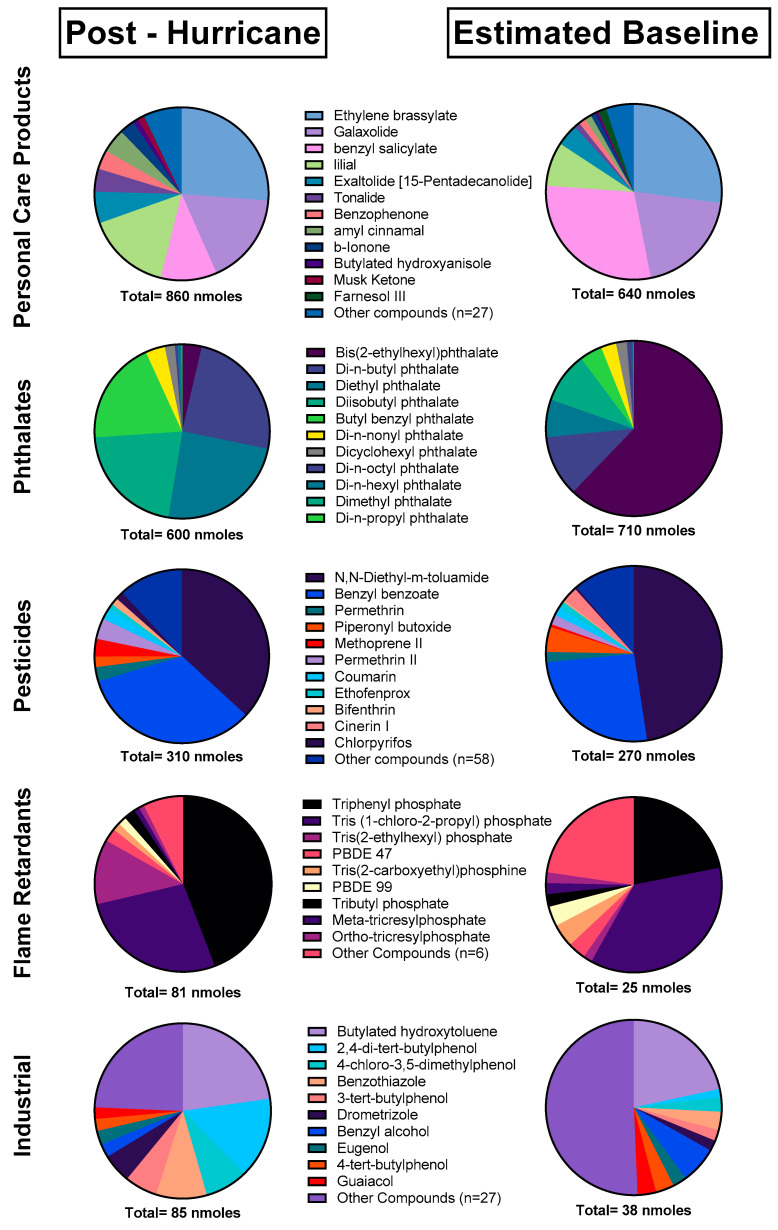
Chemical composition of chemical categories utilized in the 1500 screening method excluding PAHs. All analytes detected at least once in either timepoint were included.

**Figure 4 ijerph-19-06670-f004:**
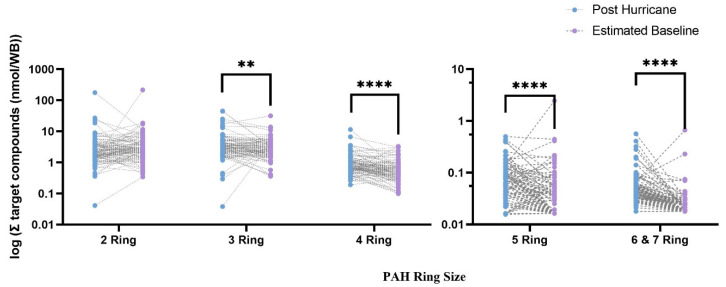
Log scale of sum concentration of PAHs across ring sizes in matched pairs across timepoints. Comparisons represent results from Wilcoxon matched-pair signed-rank tests ** (*p* < 0.01), **** (*p* < 0.0001).

**Table 1 ijerph-19-06670-t001:** Summary statistics including results from Wilcoxon matched-pairs signed rank tests for all analytes detected in the 1500 screening method detected in greater than 30% of the study population in at least one timepoint. The PAHs anthracene, 1-methylnaphthalene, naphthalene, and 2-methylphenanthrene were all excluded due to redundancy with the 63 PAH method and are discussed elsewhere. Differences were calculated by subtracting individual estimated baseline concentrations from the paired post-hurricane concentration. Analytes are ordered in decreasing overall detection frequency across both timepoints.

	Detection Frequency (%)		Mean (nmol/WB)					
Target Analyte	Post- Hurricane	Estimated Baseline	Post- Hurricane	Estimated Baseline	Average Difference	SE of Difference	Sum of Signed Rank (W)	*p* Value
Di-n-butyl phthalate	94	100	150	80	67	12	2900	<0.0001 ****
Galaxolide	95	97	150	130	21	24	1400	0.016 *
Butyl benzyl phthalate	97	98	120	30	86	11	4200	<0.0001 ****
Diisobutyl phthalate	80	99	130	67	61	12	2700	<0.0001 ****
N,N-Diethyl-m-toluamide	84	90	120	130	−11	43	1800	0.0013 **
Tonalide	85	94	37	6.7	31	5.1	3800	<0.0001 ****
Lilial	83	87	140	52	83	19	3100	<0.0001 ****
Benzophenone	86	81	31	8.4	23	3.2	4200	<0.0001 ****
Diethyl phthalate	90	81	150	50	96	18	3800	<0.0001 ****
Butylated hydroxytoluene	78	86	19	8.1	11	3.0	2700	<0.0001 ****
Tri-phenyl phosphate	86	75	36	5.4	30	6.3	4100	<0.0001 ****
Ethylene brassylate	62	86	230	170	53	44	770	0.12
Benzyl salicylate	52	93	90	190	−97	26	−1800	0.00078 ***
Di-n-nonyl phthalate	51	84	22	20	1.4	5.7	240	0.64
Amyl cinnamal	67	62	39	7.9	31	5.4	2500	<0.0001 ****
Butylated hydroxyanisole	51	67	8.6	2.2	6.5	1.0	2300	<0.0001 ****
Bis(2-ethylhexyl) phthalate	22	93	22	440	−420	85	−2900	<0.0001 ****
Benzyl benzoate	44	61	110	69	36	21	890	0.014 *
Tris(2-chloro-2-propyl) phosphate	41	72	22	9.0	13	8.1	590	0.15
b-Ionone	49	60	25	6.3	18	6.7	1600	<0.0001 ****
Permethrin	47	54	8.2	5.2	3.1	2.8	900	0.0034 **
Benzothioazole	51	39	7.9	1.3	6.7	1.2	2000	<0.0001 ****
Caffeine	39	51	58	15	43	11	1100	<0.0001 ****
Coumarin	14	57	7.1	4.5	2.6	2.1	−580	0.038 *
2,4-di-tert-butylphenol	71	12	13	0.56	12	1.1	2800	<0.0001 ****
Linalool	29	32	7.0	2.4	4.6	1.6	450	0.0058 **
Bifenthrin	35	18	3.9	0.85	3.1	1.1	630	<0.0001 ****
d-Limonene	35	18	1.9	0.36	1.5	0.61	610	<0.0001 ****
Tris(2-ethylhexyl) phosphate	42	4.0	9.7	0.41	9.3	3.1	980	<0.0001 ****

* (*p* < 0.05), ** (*p* < 0.01), *** (*p* < 0.001), **** (*p* < 0.0001).

**Table 2 ijerph-19-06670-t002:** Summary statistics including results from Wilcoxon matched-pair signed-rank tests for all detected analytes in the 63 PAH method, detected in more than 30% of the study population in at least one timepoint. Differences were calculated by subtracting individual estimated baseline concentrations from the paired post-hurricane concentration. Analytes are ordered in decreasing overall detection frequency across both timepoints.

		Detection Frequency (%)		Mean (nmol/WB)					
Ring Size	Target Analyte	Post- Hurricane	Estimated Baseline	Post- Hurricane	Estimated Baseline	Average Difference	SE of Difference	Sum of Signed Rank (W)	*p* Value
2-ring	2-Methylnaphthalene	99	100	1.17	0.693	0.476	0.621	−175	0.0021 **
	1-Methylnaphthalene	98	100	2.11	3.85	−1.74	2.25	−642	0.265
	Naphthalene	99	100	0.594	0.345	0.249	0.307	−1410	0.0132 *
	1,6 and 1,3-Dimethylnaphthalene	90	81	0.573	0.479	0.0937	0.149	−1040	0.0574
	2-Ethylnaphthalene	85	70	0.126	0.106	0.0202	0.0200	731	0.163
	1,2-Dimethylnaphthalene	70	53	0.0491	0.0509	−0.00184	0.00847	−148	0.744
	1,4-Dimethylnaphthalene	65	47	0.0448	0.0493	−0.00456	0.00971	−311	0.468
	1,5-Dimethylnaphthalene	56	43	0.0848	0.115	−0.0304	0.0210	−634	0.0948
3-ring	Retene	99	100	0.341	0.410	−0.0688	0.117	−120	0.836
	Phenanthrene	96	100	1.90	1.36	0.546	0.288	1399	0.0142 *
	1-Methylphenanthrene	93	100	0.692	0.463	0.229	0.127	882	0.125
	Dibenzothiophene	92	99	0.150	0.134	0.0161	0.0214	318	0.582
	2-Methylphenanthrene	89	100	0.645	0.556	0.0896	0.0698	388	0.501
	Fluorene	82	93	0.306	0.260	0.0456	0.0451	134	0.809
	3,6-Dimethylphenanthrene	79	76	0.0970	0.115	−0.0185	0.0137	−1140	0.0370 *
	Acenaphthene	65	9	0.240	0.0885	0.1517	0.0290	1700	<0.0001 ****
	Acenaphthylene	35	22	0.0288	0.0458	−0.0169	0.00936	−170	0.311
	Anthracene	30	19	0.401	0.0558	0.345	0.293	227	0.159
	2,3-Dimethylanthracene	39	0	0.0415	0.00117	0.0403	0.016	780	<0.0001 ****
4-ring	Pyrene	99	100	0.464	0.236	0.228	0.0449	4060	<0.0001 ****
	Fluoranthene	99	97	0.486	0.298	0.188	0.104	2590	<0.0001 ****
	1-Methylpyrene	100	93	0.0627	0.0655	−0.00288	0.00603	−644	0.263
	Triphenylene	58	31	0.0331	0.0297	0.00333	0.00540	579	0.0644
	Chrysene	57	29	0.0296	0.0286	0.000980	0.00825	671	0.0279 *
	Benz[a]anthracene	34	18	0.0200	0.0212	−0.00127	0.00615	124	0.413
5-ring	Benzo[b]fluoranthene	65	43	0.0227	0.0214	0.00136	0.00682	938	0.008 **
	Benzo[e]pyrene	61	24	0.0224	0.0159	0.006440	0.00560	1290	<0.0001 ****
	Benzo[k]fluoranthene	37	9	0.00880	0.00819	0.000619	0.00357	498	0.0003 ***
	Benzo[j]fluoranthene	30	8	0.00750	0.00767	−0.000172	0.00422	350	0.0007 ***
6-ring	Benzo[ghi]perylene	94	43	0.0218	0.00646	0.0153	0.00211	4180	<0.0001 ****
7-ring	Coronene	82	2	0.0105	0.00184	0.008690	0.000593	3320	<0.0001****

* (*p* < 0.05), ** (*p* < 0.01), *** (*p* < 0.001), **** (*p* < 0.0001).

## Data Availability

Data are available on request from the corresponding author due to, e.g., privacy or ethical restrictions.

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
