# Peer review of "Associating Increased Chemical Exposure to Hurricane Harvey in a Longitudinal Panel Using Silicone Wristbands"

_ijerph, 2022, doi:10.3390/ijerph19116670_

Round 1
Reviewer 1 Report
In my opinion the paper is well-organized and deserves to be published in a known, recognized journal. Before the paper is processed further, the authors need to acknowledge more definitely the ,,state of the art"- current meaning and the status of the research they performed so as to better stress the outcome of available literature sources. As far as I am concerned this pertains to lines 65-70 of the introduction. All other paper sections are found to be acceptable.
Author Response
Thank you. The wristband as a personal passive sampler was first published in 2014 by O’Connell et al and subsequent studies have been published by our group and other groups. In recognition of the work that has been done, we have included a recent review by Hamzai et al. (2022), which assessed 39 studies using the wristband. Following this systematic review, the authors concluded that the wristbands were successfully used to detect multiple analytes, including pesticides, flame retardants, and polycyclic aromatic hydrocarbons. The authors also characterized the stability of the wristband, noting that the ambient shipping conditions made them “especially useful for global exposure and health studies.” They also discussed the various studies that used the wristband to probe associations between concentrations found in the wristband and environmental exposures. We have briefly summarized this information in the Introduction (lines 64-71).
In summary, a growing number of studies have demonstrated positive and significantly correlated exposure biomarkers to wristbands and that wristbands have been shown to be equivalent to or better than contemporary exposure assessment tools (Hamzai et al., 2021, Waclawik et al., 2022). Silicone wristbands have been demonstrated to be an effective sampling strategy for personal exposure studies as they capture both inhalation and dermal exposure. Furthermore, we have added a citation for a recent paper from our group, highlighting the use of the personal passive sampler in disaster response research.
Thank you. The wristband as a personal passive sampler was first published in 2014 by O’Connell et al and subsequent studies have been published by our group and other groups. In recognition of the work that has been done, we have included a recent review by Hamzai et al. (2022), which assessed 39 studies using the wristband. Following this systematic review, the authors concluded that the wristbands were successfully used to detect multiple analytes, including pesticides, flame retardants, and polycyclic aromatic hydrocarbons. The authors also characterized the stability of the wristband, noting that the ambient shipping conditions made them “especially useful for global exposure and health studies.” They also discussed the various studies that used the wristband to probe associations between concentrations found in the wristband and environmental exposures. We have briefly summarized this information in the Introduction (lines 64-71). In summary, a growing number of studies have demonstrated positive and significantly correlated exposure biomarkers to wristbands and that wristbands have been shown to be equivalent to or better than contemporary exposure assessment tools (Hamzai et al., 2021, Waclawik et al., 2022). Silicone wristbands have been demonstrated to be an effective sampling strategy for personal exposure studies as they capture both inhalation and dermal exposure. Furthermore, we have added a citation for a recent paper from our group, highlighting the use of the personal passive sampler in disaster response research.
Reviewer 2 Report
This is a very interesting and well conducted study which reports, for the first time, results from individual exposure measurement from residents in areas affected by the Hurricane Harvey at two different points of time (post-hurricane and 1 year later as estimated baseline).
The manuscript has a very good introduction describing the natural-technical disasters characteristics and associated risks, and the rationale for using silicon wrist bands for measuring personal exposures post-disaster. The study design and methods are clearly described or referenced. Statistical analyses are appropriated for a panel study with inter-dependent data, and correction for multiple testing was implemented. There is no justification for the cutoff of 30% for the population detection for common chemical compounds, but the proportion is adequate to be considered a general population exposure. As recommendation a sensitivity analysis might be added using alternative cutoff values and reporting the number of elements with potential interests that were missing from the actual analysis using a different cutoff (25-35-40-45-50%).
The study used 1-year post-hurricane measurements as estimated baseline. Given that some industrial processes or population residential locations might have changed after the Hurricane, the comparison of the study findings with previous studies (before Hurricane Harvey occurred) with measurement of individual exposure in the Houston metropolitan area (groups of chemicals and maybe some specific chemicals) might add valuable information to discuss the pertinence of considering a post- disaster time as a estimated baseline.
Limitations of the interpretations of results and lack of comparability with reference chemical values or health guideline values, which are not available for many chemicals.
Author Response
We thank the reviewer for the kind words. A complete list of all analytes detected and their respective detection frequencies were included in the SI as part of our justification for having 30% as our cut-off point. Analyzing compounds with a high number of values detected below the limit of detection (LOD) can introduce bias, as indicated in the two following references [Shoari, N. and Dubé, J.-S. (2018), Toward improved analysis of concentration data: Embracing nondetects. Environ Toxicol Chem, 37: 643-656. https://doi.org/10.1002/etc.4046. Helsel, D. R. (2005). More than obvious: better methods for interpreting nondetect data. Environmental science & technology, 39(20), 419A-423A.] Regulatory agencies, including the US EPA and the European Food Safety Authority, have varying censoring points ranging from 15-to 50%. We carefully selected our cut-off of 30% following a review of other work, and we did conduct a sensitivity analysis. We tested the following cut-offs: 50%, 40%, 30%, and 15% and selected 30% to include compounds that had very different detection frequencies between the two time points (around 20% difference or more). Analysis of compound that had at least 15% detection frequency would not have yielded any additional statistically significant results for the 1500 analysis.
Regarding the reviewer’s concerns around “considering a post- disaster time as a estimated baseline,” the authors agree that a study conducted before Hurricane Harvey occurred using the same technology would have been incredibly useful. Unfortunately, those studies are not available. We discuss this as a broad limitation to all disaster response research in the discussion (lines 398-403). In order to best approximate what the baseline chemical exposure was in the Houston metropolitan area, we utilized a paired analysis to account for variability between study participants, all participants included in the study were in the same geo unit for both time points, and we sampled at the same time of the year to account for seasonal variability. The authors agree that multiple factors, including a change in industry practices, could have impacted that timepoints ability to act as a perfect baseline, which is why we refer to it as an estimated baseline.
Lastly, the authors agree that the lack of comparability to reference values at this time is a limitation, which is why we included it within the limitations section and listed it in lines 418-423. As research on SVOCs and VOCs continues, we hope that more reference values and health guidelines will be made available. There is emerging research that may enable silicone wristband concentrations to be calculated into air equivalencies, but this requires the use of performance reference compounds prior to deploying the wristband (O’Connell, S.G., Anderson, K.A. and Epstein, M.I., 2021. Determining chemical air equivalency using silicone personal monitors. Journal of exposure science & environmental epidemiology, 32(2), pp.268-279.). This study was initiated in 2017, prior to this new research. It is our hope that future research will begin to address this limitation.
Reviewer 3 Report
The article is interesting and presents a novel approach for assessing chemical exposures. I have a few specific comments listed below but my primary concern is that the findings need to be better qualified as a screening tool and that they are not representative of actual human exposure. The risk of any exposure is a function of the chemical's toxicity in relationship to its exposure. The method used does not quantify the true exposure for either the dermal or the air route, it only represents the presence of a given chemical. This information is useful but should be better qualified. Many areas should add "potential exposure" and "screening" values to be better describe the findings.
Specific comments:
1. How were blanks used and were there detectable concentrations of any of the chemicals with the blanks. This issue needs to be addressed and described in the study design.
2. Lines 72, add the year with August 25th.
3. Lines 408 and 409 - "Lastly, silicone wristbands, the sampling technology utilized in this study, capture a combination of inhalation and dermal exposure. At this time, the chemical exposures associated with a specific exposure pathway cannot be isolated". The route cannot be isolated, and the actual quantitative chemical exposure cannot be determined. Dermal and inhalation are very different routes of entry and one chemical value from a wristband doesn't provide adequate information to state anything except that the chemical is present and there is a potential for exposure to the chemical. It is not a true combination of dermal and inhalation exposure.
The information gathered is interesting. It does appear to show a difference between the two sampling times. However, it cannot be correlated directly to an exposure value for either the inhalation or the dermal route. It does provide a screening value to use for further studies of a more quantitative nature.
Also, the baseline data may not reflect seasonal variations in airborne levels and this limitation should be stated.
Author Response
Thank you for the thoughtful review of the manuscript. We have revised the manuscript to indicate that silicone wristbands do not represent total human exposure. These changes can be seen on lines 19-20, 58-60, 77-78, 110, 224, 233, 363, 380, 38, 408, and 441. Yet we also note that wristband concentrations have been shown to have better correlations to human exposure relative to hand wipes (looking at dermal exposure) (Wang, S. R., Romanak, K. A., Stubbings, W. A., Arrandale, V. H., Hendryx, M., Diamond, M. L., Salamova, A., & Venier, M. (2019). Silicone Wristbands Integrate Dermal and Inhalation Exposures to Semi-Volatile Organic Compounds (SVOCs). Environment international, 132, Article 105104. https://doi.org/10.1016/j.envint.2019.105104), active air samplers (Dixon et al., 2018), and have been shown to correlate with urinary metabolites (Dixon et al., 2018; Dixon et al., 2022). Thus, multiple studies indicate that the wristbands do measure external exposures. However, similar to hand wipes or air measurements, we have revised our language to say these measures are potentially measuring exposure.
Regarding the blanks utilized, our apologies for the confusion. The list of laboratory blanks used in the study has been added in lines 174-176. As indicated in lines 185-188 and 211-212, a laboratory processing blank or blank with the highest analyte concentration was utilized for background subtraction. A detailed account of the background subtraction applied can be found in Table SI-1.
Thank you for catching the error regarding the date on line 82. We have added in the year.
Similar to Review #2, we appreciate that Review #3 agrees with us that the inability to at this time compare wristband data to reference doses is a limitation of the study. While many of the compounds in this study do not have reference values, the available few would have added additional context. However, this does not minimize the importance of what the data means for personal exposure. We agree that the chemical exposures cannot be isolated, and we are not suggesting the chemical exposure profiles seen in the wristband are the effective dose. However, as noted above and in our response to reviewer #1, the wristband correlates well with internal biomarkers. For example, Dixon et al (2022) found that chemical concentrations in the wristband correlated well with urinary metabolites. Additionally, in response to the Reviewer #3 comment “It is not a true combination of dermal and inhalation exposure”, wristbands have been in found in some cases to better correlate with urinary metabolites than a personal active air sampler indicated that the wristbands are capturing an additional route of exposure beyond inhalation exposure (Dixon et al., 2018). Additionally, there have been similar findings when comparing wristbands, urinary metabolites, and hand wipes (Wang et al.,2019; Hammel et al., 2016). Additional background information about the wristband as a personal exposure assessment tool has been added to the introduction (lines 58-60 and 66-73), and as the reviewer suggested, the language around what the wristband data means has been changed to reflect a “potential exposure.”
Lastly, in response to the reviewers’ concern regarding seasonality and baseline data, as indicated in our response to reviewer #2, steps were taken to best approximate what conditions were like in the Houston MSA before Hurricane Harvey. We carefully timed baseline data collection to be during the same season, which was indicated in lines 134—136. We have also added this to section 4.1, lines 400-401.